# Resurrecting essential amino acid biosynthesis in mammalian cells

Julie Trolle[1†], Ross M McBee[2,3†‡], Andrew Kaufman[3], Sudarshan Pinglay[1], Henri Berger[1§], Sergei German[1], Liyuan Liu[3], Michael J Shen[1,4], Xinyi Guo[5], J Andrew Martin[1#], Michael E Pacold[6], Drew R Jones[7], Jef D Boeke[1,7,8\*], Harris H Wang[3,9\*]

[1]Institute for Systems Genetics, Department of Biochemistry and Molecular Pharmacology, NYU Langone Health, New York, United States; [2]Department of Biological Sciences, Columbia University, New York, United States; [3]Department of Systems Biology, Columbia University, New York, United States; [4]Department of Internal Medicine, NYU Langone Health, New York, United States; [5]Department of Biology, New York University, New York, United States; [6]Department of Radiation Oncology, NYU Langone Health, New York, United States; [7]Department of Biochemistry and Molecular Pharmacology, NYU Langone Health, New York, United States; [8]Department of Biomedical Engineering, NYU Tandon School of Engineering, Brooklyn, United States; [9]Department of Pathology and Cell Biology, Columbia University, New York, United States

**\*For correspondence:**
jef.boeke@nyumc.org (JDB);
hw2429@columbia.edu (HHW)

[†]These authors contributed equally to this work

**Present address:** [‡]TômTex Inc, Brooklyn, United States; [§]Weill Cornell Graduate School of Medical Sciences, New York, United States; [#]Reopen Diagnostics, LLC, Queens, United States

**Abstract** Major genomic deletions in independent eukaryotic lineages have led to repeated ancestral loss of biosynthesis pathways for nine of the twenty canonical amino acids. While the evolutionary forces driving these polyphyletic deletion events are not well understood, the consequence is that extant metazoans are unable to produce nine essential amino acids (EAAs). Previous studies have highlighted that EAA biosynthesis tends to be more energetically costly, raising the possibility that these pathways were lost from organisms with access to abundant EAAs. It is unclear whether present-day metazoans can reaccept these pathways to resurrect biosynthetic capabilities that were lost long ago or whether evolution has rendered EAA pathways incompatible with metazoan metabolism. Here, we report progress on a large-scale synthetic genomics effort to reestablish EAA biosynthetic functionality in mammalian cells. We designed codon-optimized biosynthesis pathways based on genes mined from *Escherichia coli*. These pathways were de novo synthesized in 3 kilobase chunks, assembled *in yeasto* and genomically integrated into a Chinese hamster ovary (CHO) cell line. One synthetic pathway produced valine at a sufficient level for cell viability and proliferation. $^{13}$C-tracing verified de novo biosynthesis of valine and further revealed build-up of pathway intermediate 2,3-dihydroxy-3-isovalerate. Increasing the dosage of downstream *ilvD* boosted pathway performance and allowed for long-term propagation of second-generation cells in valine-free medium at 3.2 days per doubling. This work demonstrates that mammalian metabolism is amenable to restoration of ancient core pathways, paving a path for genome-scale efforts to synthetically restore metabolic functions to the metazoan lineage.

## Editor's evaluation

In this report, the authors devised synthetic genomic strategies to introduce essential amino-acid biosynthetic pathways into mammalian cells. While the functionalization of methionine, threonine, and isoleucine synthesis was unsuccessful, restoration of valine synthesis rendered mammalian cells partially independent of exogenous valine. Moreover, transcriptomes of the valine-prototrophic

cell mirrored transcriptomes captured during recovery from valine deprivation in parental, valine-auxotrophic counterparts. Altogether, this work was found to be of substantial interest as it provides pioneering evidence that mammalian systems may be permissive to the restoration of essential amino acid biosynthetic pathways and is thus anticipated to have a broad impact in the fields of synthetic biology, biotechnology and beyond.

---

Whole genome sequencing across the tree of life has revealed the surprising observation that nine essential amino acid (EAA) biosynthesis pathways are missing from the metazoan lineage (*Payne and Loomis, 2006*). Furthermore, these losses appear to have occurred multiple times during eukaryotic evolution, including in some microbial lineages (*Figure 1A*; *Payne and Loomis, 2006*; *Guedes et al., 2011*). Branching from core metabolism, the nine EAA biosynthesis pathways missing from metazoans involve over 40 genes (*Figure 1B*, *Supplementary files 1-3*) that are widely found in bacteria, fungi, and plants (*Guedes et al., 2011*). While the absence of pathways that produce essential metabolites is observed in certain bacteria (*Zengler and Zaramela, 2018*), which possess short generation times and high genomic flexibility to adapt to rapidly changing environments, the forces driving the loss of multiple EAA biosynthetic pathways in multicellular eukaryotes remain a great mystery. An exception that proves the rule is the partial reacquisition of EAA biosynthetic pathways through horizontal gene transfer in certain rare insect lineages which host genome-reduced intracellular bacteria and feed on simple nutrient sources such as sap or blood (*Wilson and Duncan, 2015*). Recent efforts in genome-scale synthesis (*Isaacs et al., 2011*; *Mitchell et al., 2017*; *Fredens et al., 2019*) and genome-writing (*Boeke et al., 2016*) have highlighted our increasing capacity to construct synthetic genomes with novel properties, thus providing a route to not only examine these interesting evolutionary questions but also yield new capacities of bioindustrial utility (*Heng et al., 2015*; *Tan et al., 2021*; *Kitada et al., 2018*).

We sought to explore the possibility of generating prototrophic mammalian cells capable of complete biosynthesis of EAAs using a synthetic genomics approach (*Figure 1C*). The Chinese hamster ovary (CHO) K1 cell line was chosen as a model system due to its fast generation time, amenability to genetic manipulations, availability of a whole genome sequence, and established industrial relevance for producing biologics (*Fischer et al., 2015*). EAA biosynthesis genes from the best characterized model organisms were considered during pathway design while optimizing for the fewest number of enzymes needed for a given EAA pathway. To avoid using multiple promoters, we introduced ribosome-skipping 2A sequences *Szymczak-Workman et al., 2012* between biosynthetic genes to allow for protein translation of separate enzymes from a single transcriptional unit. The EAA pathway and an additional EGFP reporter were placed in a vector that could be integrated as a single copy into the CHO genome at a designated landing pad using the Flp-In system (*O'Gorman et al., 1991*). The entire pathway was synthesized de novo by commercial gene synthesis in 3 kilobase fragments and assembled in *Saccharomyces cerevisiae* via homologous recombination of 80-basepair over-laps. Subsequent antibiotic selection of cells transfected with the vector resulted in a stable cell line containing the integrated EAA pathway. Finally, we performed a variety of phenotypic, metabolomic, and transcriptomic characterizations on the modified cell line to verify activity of the EAA biosynthesis pathway.

We first confirmed that the CHO cell line was in fact auxotrophic for each of the nine EAAs. As expected, CHO-K1 did not grow in 'dropout' F-12K medium lacking each of the nine EAAs and supplemented with dialyzed fetal bovine serum (FBS) (*Figure 1—figure supplement 1*). We noted that in this cell line, canonically non-essential amino acids tyrosine and proline also exhibited EAA-like properties in dropout media. Insufficient concentrations of phenylalanine in F-12K medium or low expression of endogenous phenylalanine-4-hydroxylase that converts phenylalanine to tyrosine could underlie the tyrosine limitation. Proline auxotrophy in CHO-K1 results from epigenetic silencing of the gene encoding Δ1-pyrroline-5-carboxylate synthetase (P5CS) in the proline pathway (*Hefzi et al., 2016*). We therefore used proline as a test case for our synthetic genomics pipeline. We tested the P5CS-equivalent proline biosynthesis enzyme found in *Escherichia coli*, encoded by two separate genes, *proA* and *proB* (*Figure 1—figure supplement 2A*). A vector (pPro) carrying codon-optimized *proA*

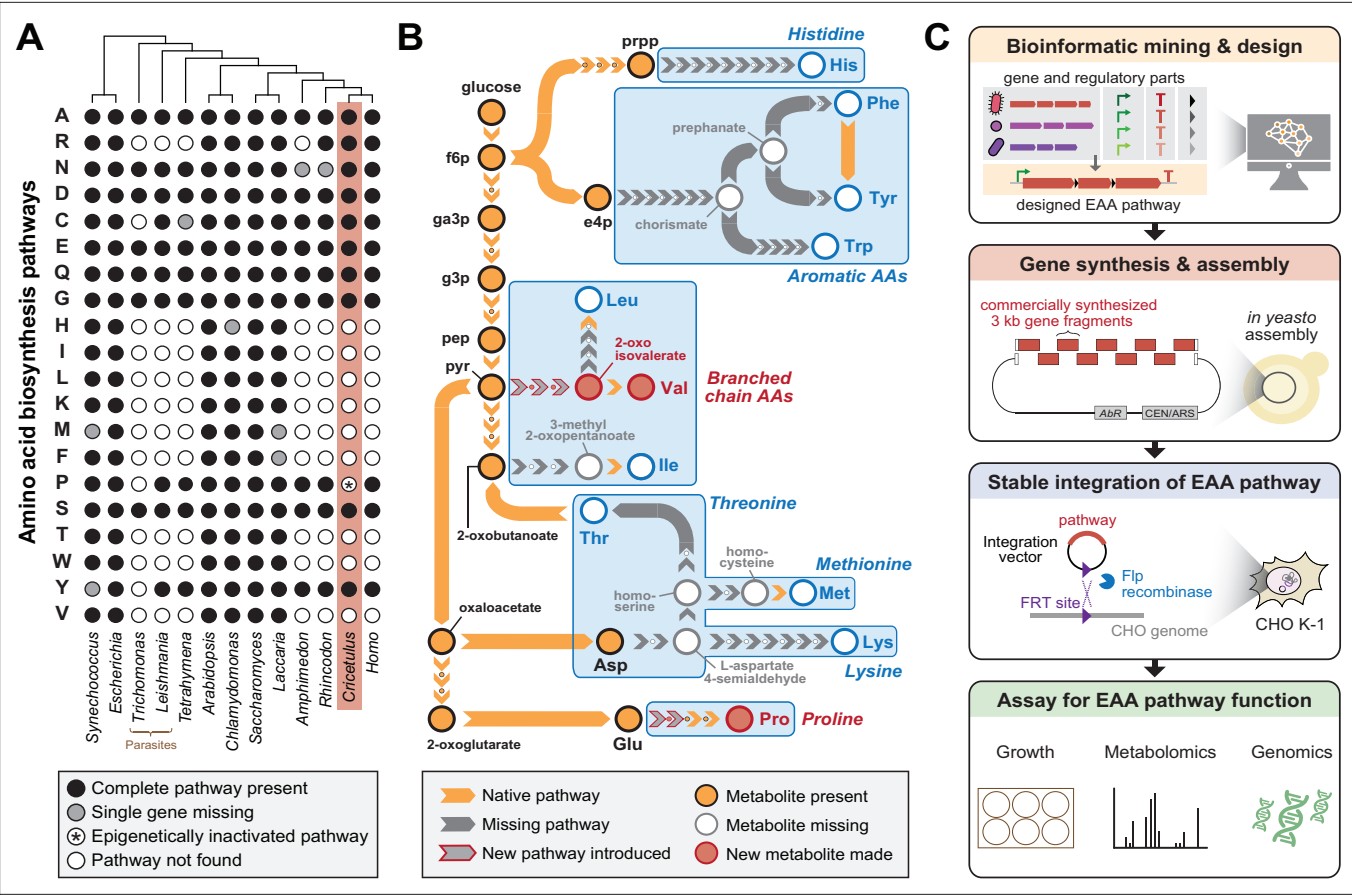

**Figure 1.** Engineering Essential Amino Acid (EAA) biosynthesis in metazoan cells. (**A**) Presence of amino acid biosynthesis pathways across representative diverse organisms on Earth. (**B**) Schematic of EAA biosynthesis pathway steps that require engineering in mammalian cells to enable complete amino acid prototrophy if imported from *Escherichia coli*. Proline and Valine pathways shown in this work are highlighted in red. (**C**) Workflow diagram of a synthetic genomics approach involving pathway design, construction, integration and testing towards mammalian EAA restoration.

The online version of this article includes the following source data and figure supplement(s) for figure 1:

**Figure supplement 1.** Amino acid dropout growth assays in CHO-K1.

**Figure supplement 1—source data 1.** Raw cell count data for amino acid dropout curves.

**Figure supplement 2.** Imported bacterial genes rescue CHO epigenetic proline auxotrophy.

**Figure supplement 2—source data 1.** Raw and processed data for proline auxotrophy rescue by *proAB* construct.

and *proB* separated by a P2A sequence was synthesized and integrated into CHO-K1 (*Figure 1— figure supplement 2B*). CHO cells with the stably integrated pPro proline pathway showed robust growth in proline-free F-12K medium (*Figure 1—figure supplement 2C-D*), thus validating a pipeline for designing and generating specific amino acid (AA) prototrophic cells.

To demonstrate restoration of EAA pathways lost from the metazoan lineage more than 650–850 million years ago (*Cunningham et al., 2017*), we built a 6-gene construct (pMTIV) to investigate the potential rescue of methionine, threonine, isoleucine, and valine auxotrophies. These EAAs were chosen because their biosynthesis pathways were missing the fewest number of genes. Valine and isoleucine collectively require four genes to recapitulate the bacterial-native pathway. (*Figure 2— figure supplement 1*). The two remaining genes were included to test potential routes to simultaneously rescue threonine and methionine auxotrophy by selectively supplementing individual missing metabolic steps, in addition to complete pathway reconstruction for valine and isoleucine. To biosynthesize methionine, we chose the *E. coli metC* gene, which encodes cystathionine-ß-lyase and converts cystathionine to homocysteine, a missing step in CHO-K1 cells in a potential serine to methionine biosynthetic pathway. Threonine production was tested using *E. coli* L-threonine aldolase *ltaE*, which converts glycine and acetaldehyde into threonine. For branched chain amino acids (BCAAs) valine and

isoleucine, three additional biosynthetic enzymes and one regulatory subunit are needed in theory to convert pyruvate and 2-oxobutanoate into valine and isoleucine, respectively. In the case of valine, pyruvate is converted to 2-acetolactate, then to 2,3-dihydroxy-isovalerate, then to 2-oxoisovalerate and finally to valine. For isoleucine, 2-oxobutanoate is converted to 2-aceto-2-hydroxybutanoate, then to 2,3-dihydroxy-3-methylpentanoate, then to 3-methyl-2-oxopentanoate, and finally to isoleucine. The final steps in the biosynthesis of both BCAAs can be performed by native CHO catabolic enzymes Bcat1 and Bcat2 *Hefzi et al., 2016*. In *E. coli*, the first three steps in the pathway are embodied in four genes that encode an acetolactate synthase with catalytic and regulatory subunits (*ilvB/N*), a ketol-acid reductoisomerase (*ilvC*), and a dihydroxy-acid dehydratase (*ilvD*) (*Figure 2A*; *Amorim Franco and Blanchard, 2017*). The final pMTIV construct comprises *metC*, *itaE*, *ilvN*, *ilvB*, *ilvC,* and *ilvD* organized as a single open reading frame (ORF) with a 2A sequence variant lying between each protein coding region (*Figure 2B*), and driven by a single strong spleen focus-forming virus (SFFV) promoter.

To test the biosynthetic capacity of pMTIV, we first introduced the construct into CHO cells. Flp-In integration was used to stably insert either pMTIV, or a control vector (pCtrl) into the CHO genome. Successful generation of each cell line was confirmed by PCR amplification of junction regions formed during vector integration (*Figure 2—figure supplement 2A-B*). RNA-seq of cells containing the pMTIV construct confirmed transcription of the entire ORF (*Figure 2B*). Western blotting of pMTIV cells using antibodies against the P2A peptide yielded bands at the expected masses of P2A-tagged proteins, confirming the production of separate distinct enzymes (*Figure 2—figure supplement 2C*).

In reconstituted methionine-free, threonine-free, or isoleucine-free F-12K medium supplemented with dialyzed FBS to reduce FBS-derived AA content (*Figure 2—figure supplement 3*), cells containing the pMTIV construct did not show viability over 7 days, similar to cells containing the pCtrl control vector (*Figure 2—figure supplement 4*). In striking contrast, however, cells containing the integrated pMTIV showed relatively healthy cell morphology and viability in valine-free F-12K medium (*Figure 2C*), whereas cells containing pCtrl exhibited substantial loss of viability over 6 days. In complete F-12K medium, cells carrying the integrated pMTIV construct showed no growth defects compared to control cells (*Figure 2D*). In valine-free F-12K medium, pMTIV cells showed a 32% increase in cell number over 6 days compared to an 88% decrease in cell number in pCtrl cells (*Figure 2E*). When cultured in valine-free F-12K medium over multiple passages with medium changes every 2 days, pMTIV cell proliferation was substantially reduced by the 3rd passage. We hypothesized that frequent passaging might over-dilute the medium and prevent sufficient accumulation of biosynthesized valine necessary for continued proliferation as has been demonstrated for certain non-essential metabolites which become essential when cells are cultured at low cell densities *Eagle and Piez, 1962*. We thus deployed a 'conditioned-medium' regimen, whereby 50% of the medium was freshly prepared valine-free F-12K medium and 50% was 'conditioned' valine-free F-12K medium in which pMTIV cells had previously been cultured over 2 days (see Materials and methods). While use of pMTIV-conditioned medium improved the survival of cells harboring the pathway, it did not completely rescue valine auxotrophy in control cells, which exhibited substantial loss of cell viability over 8 days (*Figure 2—figure supplement 5A*). As a control, we generated pCtrl-conditioned valine-free F-12K medium using the same medium conditioning regimen, which failed to enable cells to grow to the same degree as that of pMTIV-conditioned medium, suggesting that the benefit conferred by medium conditioning is valine-specific (*Figure 2—figure supplement 5B*). Using this regimen, we were able to culture pMTIV cells for 9 passages without addition of exogenous valine (*Figure 2F*). The doubling time was inconsistent across the 49 days of experimentation with cells exhibiting a mean doubling time of 5.3 days in the first 24 days and 21.0 days in the last 25 days. Despite the slowed growth seen in later passages, cells exhibited healthy morphology and continued viability at day-49, suggesting that the cells could have been passaged even further. To verify that the putative valine rescue effect was due to the valine biosynthesis genes present in pMTIV specifically and did not involve confounding effects due to the threonine and methionine biosynthesis genes included in the pMTIV pathway, we constructed and tested a second EAA pathway vector, 'pIV,' that only contained the four genes *ilvNBCD*. The pIV construct similarly supported cell growth in valine-free

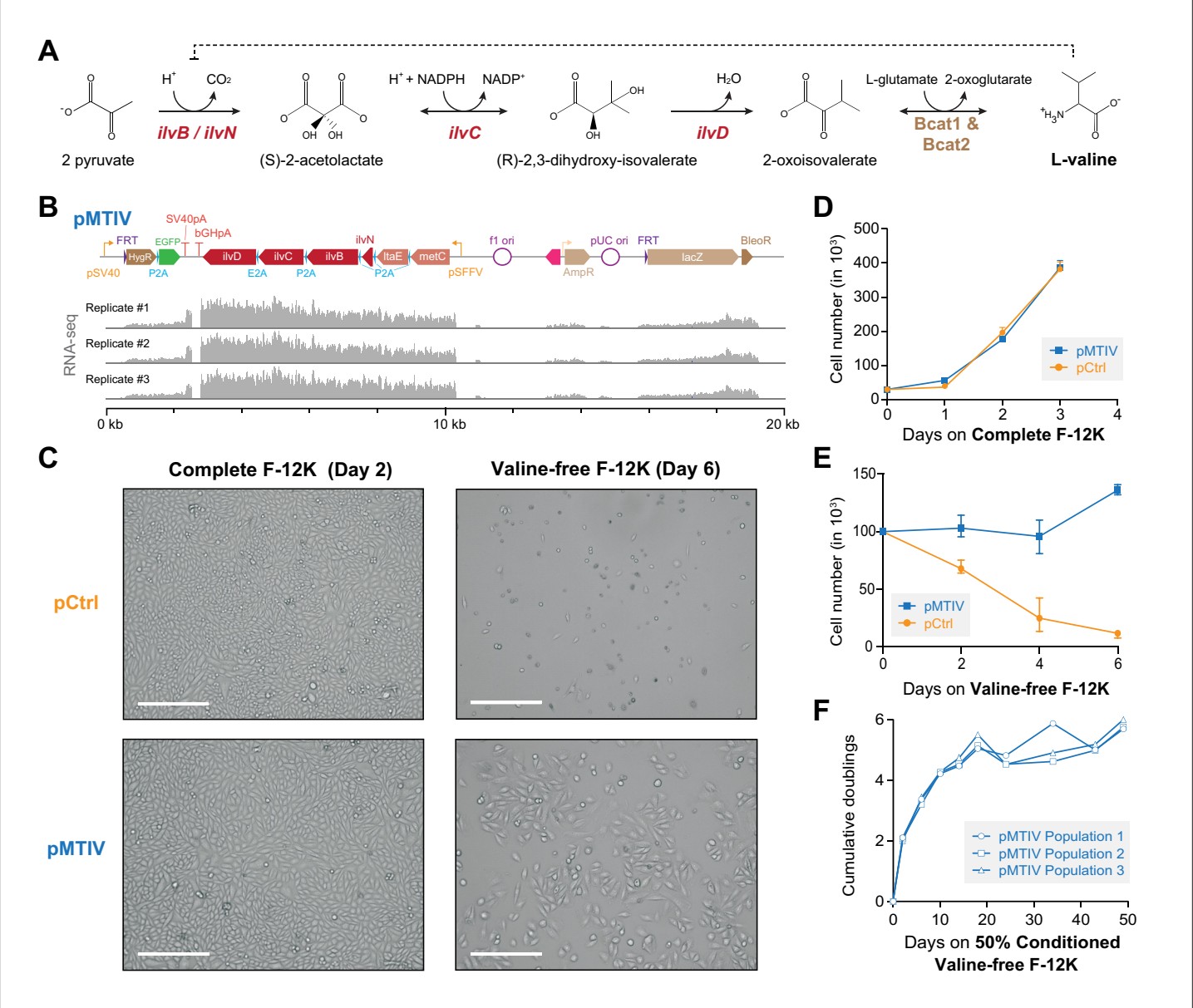

**Figure 2.** Restoration of a valine biosynthesis pathway in CHO-K1 cells. (**A**) Three enzymatic steps encoded by *E. coli* genes *ilvN* (regulatory subunit, acetolactate synthase), *ilvB* (catalytic subunit, acetolactate synthase), *ilvC* (ketol-acid reductoisomerase), and *ilvD* (dihydroxy-acid dehydratase) are required for valine biosynthesis in Chinese hamster ovary (CHO)-K1 cells. (**B**) Schematic of pMTIV construct after genomic integration and RNA-seq read coverage showing successful incorporation and active transcription. (**C**) Microscopy images of CHO-K1 cells with integrated pCtrl or pMTIV constructs in complete F-12K medium after 2 days or valine-free F-12K medium after 6 days. Scale bar represents 300 µm. (**D**) Growth curve of CHO-K1 cells with pCtrl or pMTIV in complete F-12K medium (*Figure 2—source data 1*). Day-0 indicates number of seeded cells. Error bars represent data from three replicates. (**E**) Growth curve of CHO-K1 cells with pCtrl or pMTIV in valine-free F-12K medium (*Figure 2—source data 1*). Day-0 indicates number of seeded cells. Error bars represent data from three replicates. (**F**) CHO-K1 with pMTIV cultured over nine passages in 50% conditioned valine-free F-12K medium (*Figure 2—source data 1*).

The online version of this article includes the following source data and figure supplement(s) for figure 2:

**Source data 1.** Raw cell count data for pMTIV valine-free and complete F-12K medium tests.

**Figure supplement 1.** Schematic outlines of the MTIV pathways.

**Figure supplement 2.** Confirmation of BCAA biosynthetic construct integration and 2A peptide processing by PCR and Western blot.

**Figure supplement 2—source data 1.** Raw and annotated images of assembly junction PCR gel.

**Figure supplement 2—source data 2.** Raw and annotated images of 2 A Western blot.

*Figure 2 continued on next page*

*Figure 2 continued*

**Figure supplement 3.** Amino acid content in dialyzed fetal bovine serum.

**Figure supplement 3—source data 1.** Raw and processed MS/MS data for dialyzed FBS amino acid measurements.

**Figure supplement 4.** Lack of rescue of methionine, threonine, and isoleucine auxotrophy by pMTIV construct in Chinese hamster ovary (CHO)-K1 cells.

**Figure supplement 4—source data 1.** Raw PrestoBlue assay values for MTIV construct methionine, threonine and isoleucine dropout tests *Figure 2— figure supplement 4B*.

**Figure supplement 5.** Growth behaviors on conditioned valine-free F-12K medium.

**Figure supplement 5—source data 1.** Raw cell count data for pCtrl and pMTIV cells cultured in 50% conditioned valine-free F-12K medium.

**Figure supplement 6.** The pIV construct rescues valine auxotrophy.

**Figure supplement 6—source data 1.** Raw cell count data for pIV valine-free and complete F-12K medium tests.

F-12K medium, and exhibited similar growth dynamics to the pMTIV construct in complete medium (*Figure 2—figure supplement 6*).

To confirm endogenous biosynthesis of valine, we cultured pCtrl and pMTIV cells in RPMI medium containing $^{13}C_6$-glucose in the place of its $^{12}C$ equivalent together with $^{13}C_3$-sodium pyruvate spiked in at 2 mM over three passages (*Figure 3—figure supplement 1A*). High-resolution MS1 of MTIV cell lysates revealed a peak at 123.1032 *m/z*, the expected *m/z* for $^{13}C_5$-valine (*Figure 3A*). This detected peak was subject to MS2 alongside a $^{12}C$-valine control peak and a $^{13}C_5/^{15}N$-valine peak, which was spiked into all samples to serve as an internal standard. The resulting fragmentation patterns for each peak (*Figure 3B*) matched theoretical expectations for each isotopic version of valine (*Figure 3— figure supplement 1B*). An extracted ion chromatogram further revealed a peak in the pMTIV valine-free medium metabolite extraction, which corresponded to a peak in the spiked-in positive control $^{13}C_5/^{15}N$-valine, whereas no equivalent peak was seen among metabolites extracted from pCtrl cells (*Figure 3—figure supplement 1C*). Taken together, this demonstrates that pMTIV cells are capable of biosynthesizing valine from core metabolites glucose and pyruvate, thereby proving successful metazoan biosynthesis of valine. Over the course of 3 passages in heavy valine-free medium, the non-essential amino acid alanine, which is absent from RPMI medium and synthesized from pyruvate, was found to be 86.1% $^{13}C$-labeled in pMTIV cell lysates. Assuming similar turnover rates for alanine and valine within the CHO proteome, we expected to see similar percentages of $^{13}C$-labeled valine. However, just 32.2% of valine in pMTIV cell lysates was $^{13}C$-labeled (*Figure 3—figure supplement 1D-E*). For pMTIV cells cultured in heavy but valine-replete medium, just 6.4% of valine in cell lysates was $^{13}C$-labeled. Together with the observed slow proliferation of pMTIV cells in valine-free medium, our data suggests that valine complementation is sufficient but sub-optimal for cell growth.

We performed RNA-seq to profile the transcriptional responses of cells containing pMTIV or pCtrl in complete (harvested at 0 hr) and valine-free F-12K medium (harvested at 4 hr and 48 hr, respectively) (*Figure 3C*, *Figure 3—figure supplement 2A*). The transcriptional impact of pathway integration is modest (*Figure 3D*). Only 51 transcripts were differentially expressed between pCtrl and pMTIV cells grown in complete medium, and the fold changes between conditions were small (*Figure 3E*, *Figure 3—figure supplement 2B*). While some gene ontology (GO) functional categories were enriched (*Figure 3—figure supplement 2C*), they did not suggest dramatic cellular stress. Rather, these transcriptional changes may reflect cellular response to BCAA dysregulation due to altered valine levels (*Zhenyukh et al., 2017*), or may result from cryptic effects of bacterial genes placed in a heterologous mammalian cellular context. In contrast, comparison of 48 hr valine-starved pCtrl and pMTIV cells yielded ~7500 differentially expressed genes. Transcriptomes of pMTIV cells in valine-free medium more closely resembled cells grown on complete medium than did pCtrl cells in valine-free medium (*Figure 3D*, *Figure 3—figure supplement 3A*). Differentially expressed genes between pCtrl and pMTIV cells showed enrichment for hundreds of GO categories, including clear signatures of cellular stress such as autophagy, changes to endoplasmic reticulum trafficking, and ribosome regulation (*Figure 3—figure supplement 3B*). Most of the differentially regulated genes between pCtrl cells in complete medium, and those same cells starved of valine for 48 hr were also differentially expressed when comparing pCtrl and pMTIV cells in valine-free medium (*Figure 3E*), supporting the hypothesis that most of the observed transcriptional changes represent broad but partial rescue of the cellular response to starvation. We also examined the integrated stress response (ISR) and mTOR signaling pathways, both of which are known to modulate cellular responses to starvation (*Pakos-Zebrucka*

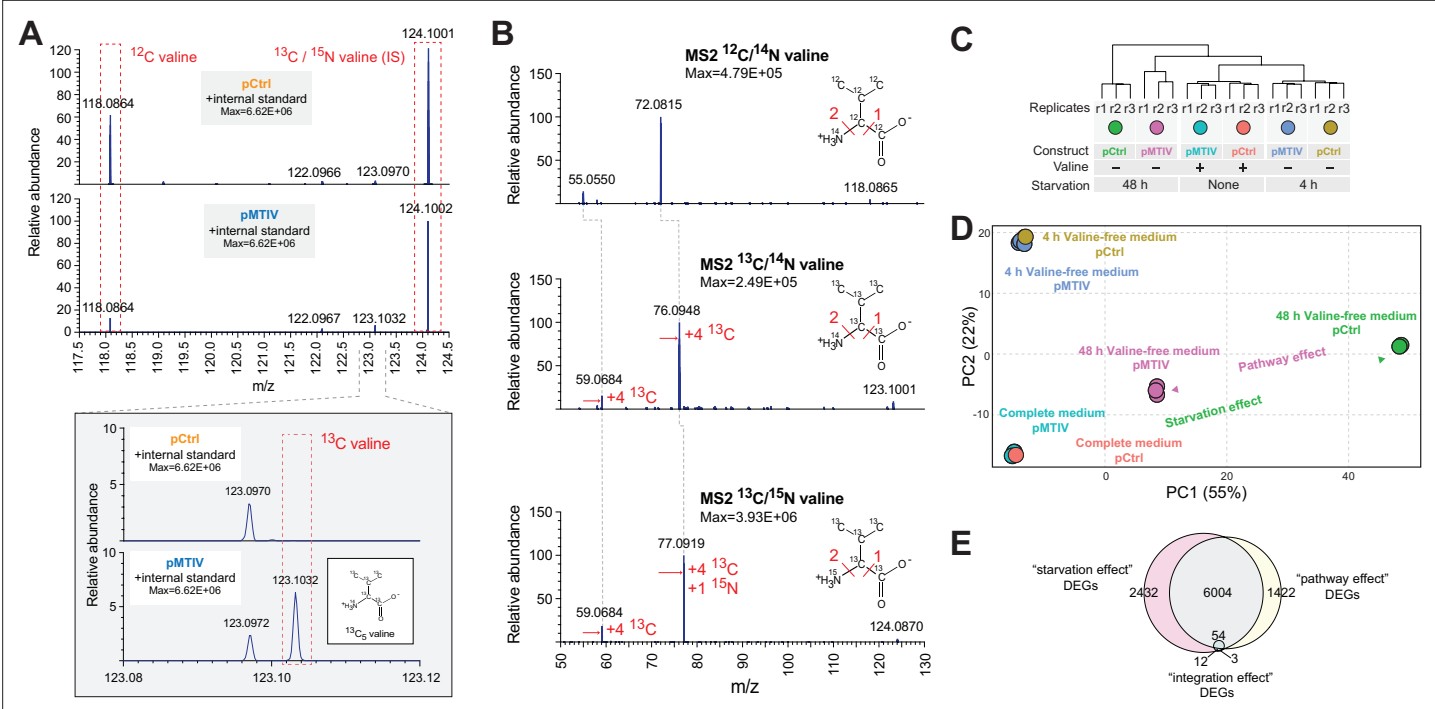

**Figure 3.** Heavy-carbon validation of endogenous valine production and transcriptomic signatures associated with rescue of nutritional starvation. (**A**) Mass spectra showing an MS1 peak corresponding to the expected *m/z* for $^{13}$C-valine when pMTIV cells were grown on valine-free RPMI medium supplemented with $^{13}$C-glucose and $^{13}$C-pyruvate, indicating autogenous production of intracellular valine (***Figure 3—source data 1***). (**B**) MS2 performed on peaks extracted from pMTIV sample, which corresponded to *m/z* values expected for $^{12}$C-valine, $^{13}$C-valine and internal standard $^{13}$C/$^{15}$N-valine. MS2 fragmentation patterns for each of these metabolites matched expectations (***Figure 3—source data 1***). (**C**) RNA-seq dendrogram of pCtrl cells and pMTIV cells grown on complete F-12K medium or starved of valine for 4 hr or 48 hr. (**D**) Principal Component Analysis (PCA) space depiction of pCtrl cells and pMTIV cells grown on complete F-12K medium, or starved of valine for 4 hr or 48 hr. (**E**) Overlap between differently expressed genes (DEGs) comparing pCtrl cells cultured in valine-free F-12K medium to pCtrl cells cultured in complete F-12K medium (the 'starvation effect'), pCtrl cells and pMTIV cells after 48 hr starvation on valine-free F-12K medium (the 'pathway effect'), and pCtrl cells and pMTIV cells grown on complete F-12K medium (the 'integration effect').

The online version of this article includes the following source data and figure supplement(s) for figure 3:

**Source data 1.** Raw MS/MS data for $^{13}$C valine labeling experiments.

**Figure supplement 1.** Heavy-labeled metabolomics confirms endogenous valine production.

**Figure supplement 1—source data 1.** Further raw MS/MS data for $^{13}$C valine labeling experiments.

**Figure supplement 2.** Transcriptomic analysis of pMTIV pathway effect in complete F-12K medium.

**Figure supplement 3.** Transcriptomic analysis of pMTIV pathway effect in valine-free F-12K medium.

**Figure supplement 4.** Transcriptomic analysis of pMTIV pathway effect on mTOR and the integrated stress response.

**Figure supplement 5.** Transcriptomic analysis of long-term (29 days) pMTIV pathway effect in conditioned valine-free F-12K medium.

*et al., 2016*). We observed no clear signatures of mTOR activation (***Figure 3—figure supplement 4A***), although a number of individual genes related to the mTOR pathway were significantly differentially expressed compared to pCtrl cells valine-starved for 48 hr (***Figure 3—figure supplement 4B***). A manually curated list of ISR genes showed signals of ISR gene activation, but showed few differences between pCtrl and pMTIV cells at 48 hr of starvation (***Figure 3—figure supplement 4C***). pMTIV cells grown for 5 passages over 29 days on conditioned valine-free F-12K medium were more similar to pMTIV cells starved for 48 hr than to pCtrl cells starved for 48 hr (***Figure 3—figure supplement 5A***). The transcriptomic signature of these cells, appears slightly orthogonal to the 'starvation' effect and

recovery of pMTIV and pCtrl cells starved for 48 hr in PCA space (*Figure 3—figure supplement 5B*), but the majority of the differently expressed genes between pMTIV cells starved for 29 days compared to pCtrl cells grown on complete media are shared with cells starved for 48 hr (*Figure 3—figure supplement 5C*), suggesting some limited cellular adaptation to long term passaging on valine-free media.

To improve rescue of the valine starvation phenotype, we looked for valine biosynthetic pathway intermediates in our metabolomics data that might suggest that the pathway was bottlenecked at any stage. While no signal could be detected for pyruvate, 2-acetolactate or 2-oxoisovalerate, a signal was detected for pathway intermediate $^{13}C_5$-2,3-dihydroxy-isovalerate, which was specific to pMTIV cells cultured in both complete and in valine-free medium (*Figure 3—figure supplement 1F*). To determine whether the downstream pathway gene, *ilvD*, which encodes the dihydroxy-acid dehydratase enzyme, might constitute a bottleneck, we generated a lentivirus encoding a puromycin resistance cassette in addition to *ilvD* under control of a viral MMLV promoter (*Figure 4A*). Both pCtrl and pMTIV cells were infected and integrants were selected for on puromycin, resulting in a population-averaged integration count of 5.7 copies of *ilvD* for pCtrl cells and 7.1 for pMTIV cells. This resulted in a 0.27- and 0.54-fold increase in *ilvD* expression (measured as a proportion of uninfected pMTIV *ilvD* expression) for infected pCtrl (pCtrl-ilvD+) and infected pMTIV cells (pMTIV-ilvD+), respectively (*Figure 4B*).

We characterized pMTIV and pMTIV-ilvD + cells by culturing these in isotopically heavy valine-free RPMI medium containing $^{13}C_6$-glucose as before; however, in this experiment 2 mM $^{12}C$ sodium pyruvate was spiked into the medium rather than $^{13}C$-sodium pyruvate (*Figure 4—figure supplement 1A*). We also reduced the isoleucine content of this isotopically heavy valine-free RPMI medium to match the isoleucine content of F-12K medium from 0.38 mM to 0.06 mM after observing that pMTIV cells proliferated faster in this condition, presumably due to the high sensitivity of the *ilvN/B*-encoded acetolactate synthase complex to isoleucine-induced negative feedback inhibition (*Barak et al., 1987*; *Figure 4—figure supplement 1B*). pMTIV and pMTIV-ilvD + cells were cultured in this reduced-isoleucine heavy medium over 36 days on plates coated with 0.1% gelatin (*Figure 4—figure supplement 1C*). We collected duplicate samples at 6 and 10 different timepoints for pMTIV and pMTIV-ilvD+, respectively, throughout the 36 days of culture. Metabolomics analysis revealed $^{13}C$-valine labeling with most of the valine in both pMTIV and pMTIV-ilvD + cells carrying 0, 2, 3, or 5 $^{13}C$ carbons at both an early (2 days) and a late (24 days) time point in accordance with the expectation when both $^{12}C_3$ and $^{13}C_3$-pyruvate (derived from $^{13}C_6$-glucose) are present (*Figure 4C*). In pMTIV cells, presence of pathway intermediate 2,3-dihydroxy-isovalerate fluctuated throughout the 24 days of culture with cells exhibiting higher concentrations in earlier time points. In contrast, little or no 2,3-dihydroxy-isovalerate was detected in pMTIV-ilvD + cells across the entire time course, suggesting that the introduction of the extra copies of *ilvD* is addressing a pathway bottleneck as hypothesized (*Figure 4D*). In pMTIV-ilvD + cells, the percentage of valine carrying at least one $^{13}C$-labeled carbon also remained relatively steady throughout the 36 days, while pMTIV cells again exhibited more variability (*Figure 4—figure supplement 1D*). We calculated the molar content of valine within the cells by relativizing peaks to that of the $^{13}C/^{15}N$-valine internal standard, which was spiked into each sample at a known concentration. pMTIV samples on average contained 352.0 picomoles of valine per million cells of which 212.5 picomoles was $^{13}C$-labeled whereas pMTIV-ilvD + samples on average contained 423.0 picomoles of valine per million cells of which 242.5 picomoles was $^{13}C$-labeled (*Figure 4E*). Collectively, the stabilization of 2,3-dihydroxy-isovalerate and valine levels in pMTIV-ilvD + cells suggests that the introduction of additional *ilvD*-encoded dihydroxy-acid dehydratases is indeed improving flux through the pathway, allowing cells to respond to low valine levels more efficiently compared to pMTIV cells, which results in a more homeostatic cell state.

We quantified the functional impact of modifying flux at this pathway bottleneck by culturing both cell lines in unconditioned, reduced-isoleucine, valine-free RPMI medium containing 2 mM sodium pyruvate over 10 passages on plates not coated with gelatin. When cultured in this medium, pCtrl-ilvD + cells displayed rapid loss of cell viability similar to pCtrl cells (*Figure 5—figure supplement 1A*). However, pMTIV-ilvD + cells displayed an improved growth phenotype compared to pMTIV cells

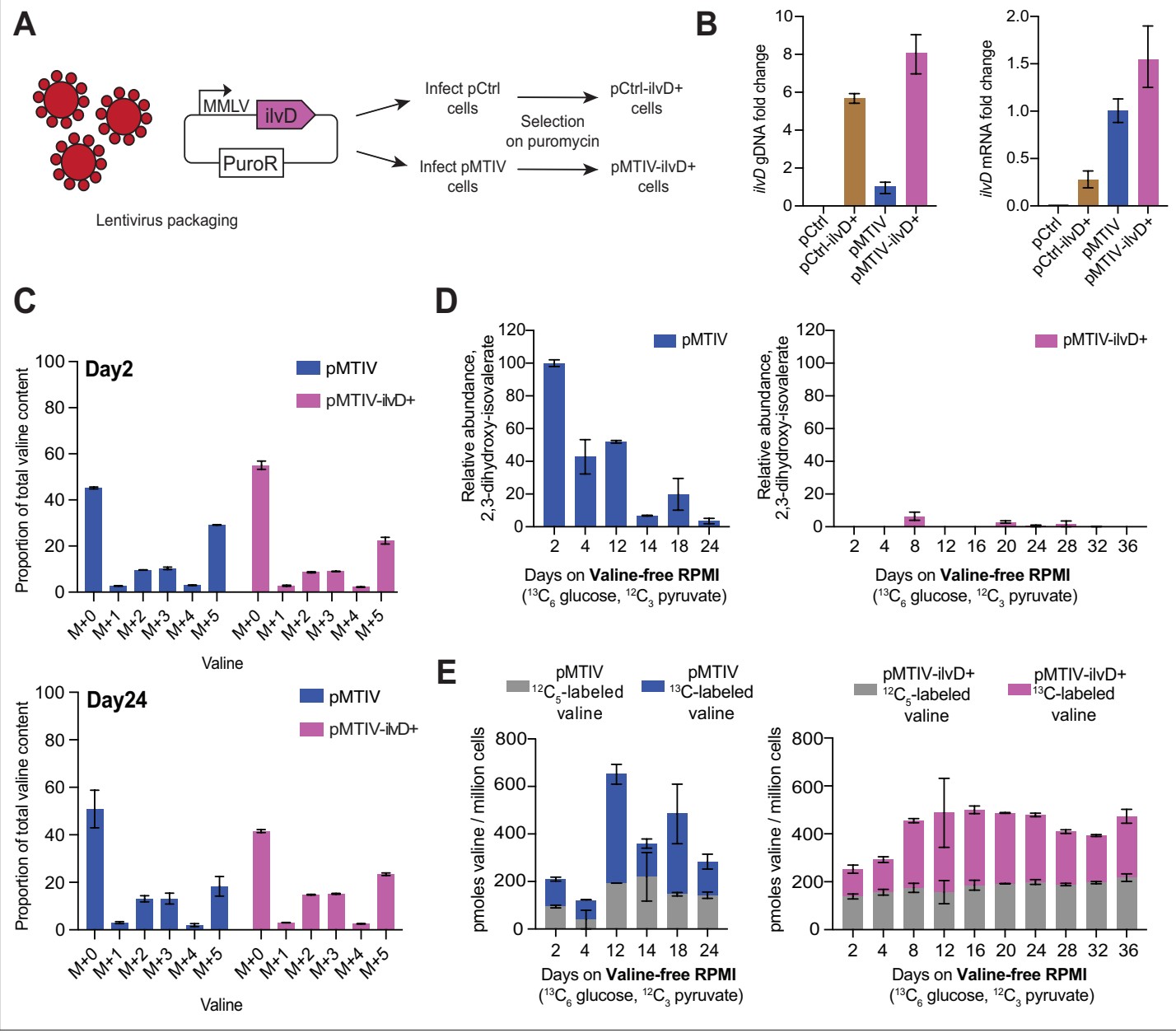

**Figure 4.** Optimization of prototrophic functionality reduces pathway bottlenecking and stabilizes valine availability at late time points in valine-free conditions. (**A**) Infection of pCtrl and pMTIV cells using a lentivirus carrying extra copies of *ilvD* led to the generation of the pCtrl-ilvD + and pMTIV-ilvD + cell lines. (**B**) *ilvD* qPCR on gDNA and cDNA from each cell line (***Figure 4—source data 1***). Fold change levels were relativized to pMTIV. cDNA was reverse transcribed using oligo(dT) primers from RNA templates collected from each cell line. Error bars show SD of three technical replicates. (**C**) $^{13}$C-labeling of valine in pMTIV and pMTIV-ilvD + samples collected on day-2 and day-24 of culture in unconditioned, reduced-isoleucine (0.06 mM), valine-free RPMI medium containing $^{13}$C$_6$-glucose with 2 mM $^{12}$C-sodium pyruvate spiked-in on plates coated with 0.1% gelatin (***Figure 4—source data 1***). Error bars represent data from two replicates. (**D**) Relative abundance of pathway intermediate 2,3-dihydroxy-isovalerate in pMTIV and pMTIV-ilvD + cells cultured as in (**C**) (***Figure 4—source data 1***). Error bars represent data from two replicates. (**E**) Molar concentration of valine in pMTIV and pMTIV-ilvD + cells cultured as in (**C**) (***Figure 4—source data 1***). $^{12}$C-labeled refers to valine carrying $^{12}$C exclusively while $^{13}$C-labeled refers to valine carrying at least one $^{13}$C. Error bars represent data from two replicates.

The online version of this article includes the following source data and figure supplement(s) for figure 4:

*Figure 4 continued on next page*

*Figure 4 continued*

**Source data 1.** Control and ilvD qPCR data for pMTIV and pMTIV-ilvD + cells, raw cell count data for pMTIV cultured in valine-free RPMI with different BCAA concentrations, and MS/MS data for long-term pMTIV and pMTIV-ilvD+ [13]C valine labeling experiments.

**Figure supplement 1.** Long-term culturing of cells in unconditioned isotopically heavy valine-free RPMI medium.

**Figure supplement 1—source data 1.** Growth curve and MS/MS data for long-term pMTIV and pMTIV-ilvD+ [13]C valine labeling experiments.

with an average doubling time of 3.2 days across 39 days and an apparent ability to proliferate indefinitely (*Figure 5*). By comparison, pMTIV cells exhibited an average doubling time of 4.3 days across 19 days after which they displayed loss of cell viability following passaging events despite both cell lines exhibiting comparable doubling times in complete medium (*Figure 5—figure supplement 1B*).

In this work, we demonstrated the successful restoration of an EAA biosynthetic pathway in a metazoan cell. Our results indicate that contemporary metazoan biochemistry can support complete biosynthesis of valine, despite millions of years of evolution from its initial loss from the ancestral lineage. Interestingly, independent evidence for BCAA biosynthesis has also been obtained for sap-feeding whitefly bacteriocytes that host bacterial endosymbionts; metabolite sharing between these cells is predicted to lead to biosynthesis of BCAAs that are limiting in their restricted diet. The malleability of mammalian metabolism to accept heterologous core pathways opens up the possibility of animals with designer metabolisms and enhanced capacities to thrive under environmental stress and nutritional starvation (*Zhang et al., 2014*). Yet, our failure to functionalize designed methionine, threonine, and isoleucine pathways highlights outstanding challenges and future directions for synthetic metabolism engineering in animal cells and animals. Other pathway components or alternative selections may be needed for different EAAs (*Rees and Hay, 1995*). For instance, the *ilvB/N* enzyme pair we chose to encode the acetolactate synthase complex in our construct shows feedback sensitivity to isoleucine, which may account for the failure of the pathway to rescue isoleucine auxotrophy. A general lack of predictability and a dearth of well-characterized and controllable genetic 'parts' with high dynamic range continue to hamper efforts in genome-scale mammalian engineering (*Black et al., 2017*; *Gilbert et al., 2014*; *Weingarten-Gabbay et al., 2019*). Studies to reincorporate EAAs into the core mammalian metabolism could provide greater understanding of nutrient-starvation in different physiological contexts including the tumor microenvironment (*Lim et al., 2020*), help answer deep evolutionary questions regarding the formation of the metazoan lineage (*Tan et al., 2009*), and lead to new model systems or even therapeutics to address metabolic syndrome, Maple Syrup Urine Disease *Dancis et al., 1960* and Phenylketonuria (*Blau et al., 2010*), all of which involve amino acid biosynthetic dysfunction (*Kemmer et al., 2010*; *Ye et al., 2013*). Emerging synthetic genomic efforts to build a prototrophic mammal may require reactivation of many more genes (*Supplementary files 1-3*), iterations of the

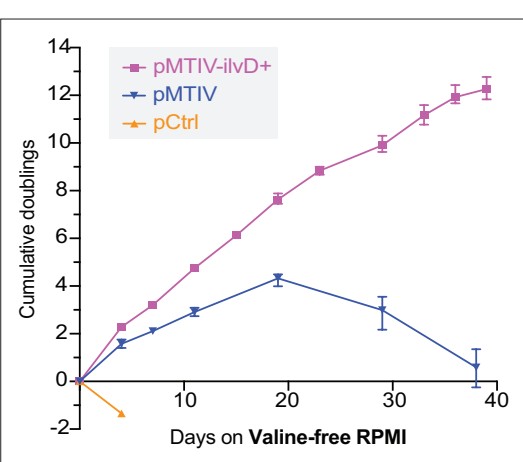

**Figure 5.** Optimized prototrophic cells double every 3.2 days in unconditioned valine-free RPMI medium over 39 days. Population doubling level of pCtrl, pMTIV, pMTIV-ilvD + cells cultured in unconditioned, reduced-isoleucine, valine-free RPMI medium containing 2 mM sodium pyruvate (*Figure 5—source data 1*). Plates were not coated with gelatin. Error bars represent data from three replicates.

The online version of this article includes the following source data and figure supplement(s) for figure 5:

**Source data 1.** Raw and processed cell count data for pMTIV and pMTIV-ilvD + cultured long-term in unconditioned valine-free RPMI medium.

**Figure supplement 1.** Comparing growth of cell lines with elevated *ilvD* copy numbers to those without.

**Figure supplement 1—source data 1.** Raw cell count data for pCtrl, pMTIV, pCtrl-ilvD + and pMTIV-ilvD + cultured in valine-free and complete RPMI medium.

design, build, test (DBT) cycle, and a larger coordinated research effort to ultimately bring such a project to fruition.

# Materials and methods

**Key resources table**

| Reagent type (species) or resource | Designation | Source or reference | Identifiers | Additional information |
|---|---|---|---|---|
| Strain, strain background (*E. coli*) | TransforMax EPI300 | Lucigen | C300C105 | Chemically competent |
| Strain, strain background (*E. coli*) | NEB 10-beta | New England Biolabs | C3020K | Electrocompetent |
| Cell line (C. griseus) | CHO Flp-In | ThermoFisher | R75807 | |
| Cell line (C. griseus) | CHO pMTIV | This paper | | Cell line maintained in J.D.Boeke lab |
| Cell line (C. griseus) | CHO pCtrl | This paper | | Cell line maintained in J.D.Boeke lab |
| Cell line (C. griseus) | CHO pMTIV-ilvD+ | This paper | | Cell line maintained in J.D.Boeke lab |
| Cell line (C. griseus) | CHO pCtrl-ilvD+ | This paper | | Cell line maintained in J.D.Boeke lab |
| Cell line (C. griseus) | CHO pPro | This paper | | Cell line maintained in H.H.Wang lab |
| Cell line (C. griseus) | CHO pCtrl-mCh | This paper | | Cell line maintained in H.H.Wang lab |
| Recombinant DNA reagent | pJTR0381 (plasmid) | This paper | | Encodes Ec *ilvD* transcription units for lentiviral transduction |
| Antibody | Anti-2A peptide (mouse monoclonal) | Novus Biologicals | NBP2-59627 | 1:1000 dilution |
| Antibody | Anti-α/β-tubulin (rabbit polyclonal) | Cell Signaling Technology | 2148 | 1:1000 dilution |
| Antibody | IRDye 800CW Anti-mouse IgG (goat polyclonal) | LI-COR | 926–32210 | 1:20000 dilution |
| Antibody | IRDye 680 RD Anti-rabbit IgG (goat polyclonal) | LI-COR | 926–68071 | 1:20000 dilution |
| Commercial assay or kit | PrestoBlue Cell Viability Reagent | ThermoFisher | A13261 | |
| Commercial assay or kit | NEBNext poly(A) mRNA Magnetic Isolation module | New England Biolabs | E7490 | |
| Commercial assay or kit | NEBNext Ultra RNA Library Prep Kit for Illumina | New England Biolabs | E7770 | |
| Other | F-12K medium without amino acids, powder base | US Biological | N8545 | Dropout medium base |
| Other | RPMI 1640 without amino acids & glucose, powder base | US Biological | R9010-01 | Dropout medium base |
| Other | Gibco fetal bovine serum, dialyzed | Fisher Scientific | 26400044 | Lot 2140178 |

CHO: Chinese hamster ovary.

## Pathway completeness analysis

For pathway completeness analysis, the EC numbers of each enzyme in each amino acid biosynthesis pathway (excluding pathways annotated as only occurring in prokaryotes) were collected from the MetaCyc database (*Supplementary file 4*). Variant biosynthetic routes to the same amino acid were considered as separate pathways, generating distinct EC number lists. The resulting per-pathway EC number lists were checked against the KEGG, Entrez Gene, Entrez Nucleotide, and Uniprot databases using their respective web APIs for each listed organism. If the combination of all databases contained at least one complete EC numbers list, corresponding to an end-to-end complete biosynthetic pathway, the organism was considered 'complete' for that essential amino acid.

## Cell lines and media

CHO Flp-In cells (ThermoFisher, R75807) were used in all experiments. All cell lines tested negative for mycoplasma. For growth assays involving amino acid dropout formulations, medium was prepared from an amino acid-free Ham's F-12 (Kaighn's) powder base (US Biological, N8545), and custom combinations of amino acids were added back in as needed to match the standard amino acid concentrations for Ham's F-12 (Kaighn's) medium or as specified. Custom amino acid dropout medium was adjusted to a pH of 7.3, sterile filtered, and supplemented with 10% dialyzed Fetal Bovine Serum (Fisher Scientific, 26400044) and Penicillin-Streptomycin (100 U/mL) prior to use. For metabolomics experiments, medium was prepared from an amino acid-free and glucose-free RPMI 1640 powder base (US Biological, R9010-01), and custom combinations of amino acids and isotopically heavy glucose and sodium pyruvate were added in to match the standard amino acid concentrations for RPMI 1640 or as specified. pH was adjusted to 7.3, sterile filtered and supplemented with 10% dialyzed Fetal Bovine Serum and Penicillin-Streptomycin (100 U/mL) prior to use. Where specified, cells were cultured on plates coated with 0.1% gelatin (EMD Millipore, ES-006-B) for 30 min at room temperature. Plates were washed with PBS prior to use.

## Cell counting and quantification

For evaluating effects of amino acid dropout on cell growth curves, cells were seeded at $1\times10^4$ into 6-well plates into F12-K media with lowered amino acid concentrations relative to typical F12-K media and then allowed to grow for 5 days. Media was then aspirated off and replaced with PBS with Hoechst 33342 live nuclear stain for automated imaging and counting using a DAPI filter set on an Eclipse Ti2 automated inverted microscope. To count, an automated microscopy routine was used to *Figure 5* random locations within each well at 10× magnification, and then the cells present in imaged frames counted using automatic cell segregation and counting software. Given differences in cell response to starvation, segregation and counting parameters were tuned in each experiment, but kept constant between starvation conditions and cells with and without the pathway. For synthetic prototroph pathway tests, raw cell counts were performed using the Countess II Automated Cell Counter (ThermoFisher, A27977) in accordance with the manufacturer's protocol, or using the Scepter 2.0 Handheld Automated Cell Counter (Millipore Sigma, C85360) in accordance with the manufacturer's protocol. Where indicated, relative cell quantification was measured using PrestoBlue Cell Viability Reagent (ThermoFisher, A13261) in accordance with the manufacturer's protocol.

## Cell culture in conditioned medium

Conditioned medium was generated by seeding $1\times10^6$ pMTIV cells into 10 mL complete F12-K medium on 10 cm plates and replacing the medium with 10 mL freshly prepared valine-free F12-K medium the next day following a PBS wash step. Cells conditioned the medium for 2 days at which point the medium was collected, centrifuged at 300×g for 3 mins to remove potential cell debris, sterile filtered, and collected in 150 mL vats to reduce batch-to-batch variation. This 100% conditioned medium was subsequently mixed in a 1:1 ratio with freshly prepared, unconditioned valine-free medium to generate so-called 50% conditioned valine-free medium, which was used throughout the long-term culturing process of synthetic prototrophic cells without exogenous supply of valine where specified.

## DNA assembly, recovery and amplification

Integrated constructs were synthesized de novo in 3 kb DNA segments with each segment overlapping neighboring segments by 80 bp. Assembly was conducted *in yeasto* by co-transformation of segments into *S. cerevisiae* strain BY4741 made competent by the LiOAc method (*Pan et al., 2007*). After 2 days of selection at 30°C on SC–Ura medium, individual colonies were picked and cultured overnight. 1.5 mL of each resulting yeast culture was resuspended in 250 µl P1 resuspension buffer (Qiagen, 19051) containing RNase. Glass beads were added to each resuspension and the mixture was vortexed for 10 mins to mechanically shear the cells. Next, cells were subject to alkaline lysis by adding 250 µl of P2 lysis buffer (Qiagen, 19052) for 5 mins and then neutralized by addition of Qiagen N3 neutralization buffer (Qiagen, 19053). Subsequently, cell debris was spun down and plasmid DNA was collected using the Zymo Zyppy plasmid preparation kit (Zymo Research, D4036) according to the manufacturer's instructions. Plasmid DNA was eluted in Zyppy Elution buffer and subsequently

transformed into TransforMax EPI300 chemically competent *E. coli* in accordance with the manufacturer's protocols (Lucigen, C300C105) for plasmid amplification.

## Protein extraction and western blot

Cell were lysed in SKL Triton lysis buffer (50 mM Hepes pH7.5, 150 mM NaCl, 1 mM EDTA, 1 mM EGTA, 10% glycerol, 1% Triton X-100, 25 mM NaF, 10 µM $ZnCl_2$) supplemented with protease inhibitor (Sigma 11873580001). NuPAGE LDS sample buffer (ThermoFisher, NP0007) supplemented with 1.43 M β-mercaptoethanol was added to samples prior to heating at 70°C for 10 mins. Gel electrophoresis was performed using 4–12% Bis-Tris gels (ThermoFisher, NP0326BOX) and run in NuPAGE MOPS running buffer (ThermoFisher, NP0001). Proteins were then transferred onto a PVDF membrane (Milipore Sigma, IPFL00010) using the Biorad Trans-Blot Turbo system in accordance with the manufacturer's instructions. The transfer membrane was blocked in Odyssey blocking buffer (LI-COR, 927–40000) for 1 hr at room temperature prior to incubation in primary antibody (Novus Biologicals, NBP2-59627 [1:1000 dilution]; Cell Signaling Technology, 2148 [1:1000 dilution]) solubilized in a 1:1 mixture of Odyssey blocking buffer and TBS-T buffer (50 mM Tris Base, 154 mM NaCl, 0.1% Tween20) overnight at 4°C. Secondary antibodies (LI-COR, 926–32210 [1:20,000 dilution]; LI-COR, 926–68071 [1:20,000 dilution]),were also solubilized in Odyssey blocking buffer / TBS-T buffer. The membrane was incubated in the secondary antibody solution for 1.5 hr at room temperature.

## Metabolomics

Cells were cultured in RPMI medium containing $^{13}C$-glucose and/or $^{13}C$-sodium pyruvate were indicated prior to cell harvest. Cell pellets were generated by trypsinization, followed by low speed centrifugation, and the pellet was frozen at –80°C until further processing. A metabolite extraction was carried out on each sample with an extraction ratio of 1e6 cells per mL (80% methanol containing internal standards, 500 nM), according to a previously described method (*Pacold et al., 2016*). The LC column was a Millipore ZIC-pHILIC (2.1×150 mm, 5 µm) coupled to a Dionex Ultimate 3000 system and the column oven temperature was set to 25°C for the gradient elution. A flow rate of 100 µL/min was used with the following buffers; (A) 10 mM ammonium carbonate in water, pH 9.0, and (B) neat acetonitrile. The gradient profile was as follows; 80–20%B (0–30 min), 20–80%B (30–31 min), 80–80%B (31–42 min). Injection volume was set to 1 µL for all analyses (42 min total run time per injection). MS analyses were carried out by coupling the LC system to a Thermo Q Exactive HF mass spectrometer operating in heated electrospray ionization mode (HESI). Method duration was 30 min with a polarity switching data-dependent Top 3 method for both positive and negative modes, and targeted MS2 scans for the monoisotopic, U-$^{13}C$, and U-$^{13}C$/U-$^{15}N$ valine *m/z* values. Spray voltage for both positive and negative modes was 3.5 kV and capillary temperature was set to 320°C with a sheath gas rate of 35, aux gas of 10, and max spray current of 100 µA. The full MS scan for both polarities utilized 120,000 resolution with an AGC target of 3e6 and a maximum IT of 100 ms, and the scan range was from 67 to 1000 *m/z*. Tandem MS spectra for both positive and negative mode used a resolution of 15,000, AGC target of 1e5, maximum IT of 50ms, isolation window of 0.4 m/z, isolation offset of 0.1 m/z, fixed first mass of 50 m/z, and 3-way multiplexed normalized collision energies (nCE) of 10, 35, 80. The minimum AGC target was 1e4 with an intensity threshold of 2e5. All data were acquired in profile mode. All valine data were processed using Thermo XCalibur Qualbrowser for manual inspection and annotation of the resulting spectra and peak heights referring to authentic valine standards and labeled internal standards as described.

## RNA Seq

RNA was extracted from cells using the Qiagen RNeasy Kit (Qiagen, 74104) according to the manufacturer's protocol. QIAshredder homogenizer columns (Qiagen, 79654) were used to disrupt the cell lysates. mRNA was purified using the NEBNext poly(A) mRNA Magnetic Isolation module (New England Biolabs, E7490) in accordance with the manufacturer's protocol. Libraries were prepared using the NEBNext Ultra RNA Library Prep Kit for Illumina (New England Biolabs, E7770), and sequenced on a NextSeq 550 single-end 75 cycles high output with v2.5 chemistry. Reads were adapter and quality trimmed with fastP using default parameters and psuedoaligned to the GCF_003668045.1_CriGri-PICR Chinese hamster genome assembly using kallisto. Differential gene enrichment analysis was performed with in R with DESeq2 and GO enrichment performed and visualized with clusterProfiler

against the org.Mm.eg.db database, with further visualization with the pathview, GoSemSim, eulerr packages. Differentially expressed were considered to be statistically significant if their DEseq2 corrected expression was than or equal to the package default of p≤0.05. GO categories were considered to be statistically significantly regulated if their de/enrichment was calculated as a multiple testing corrected value of p≤0.05. For mTOR-related genes, all genes present in all GO categories containing the keyword 'TOR' were tested for significance with a multiple testing corrected value of p≤0.05. For ISR genes, a manually curated list of genes consisting of *Eif2ak4, Eif2ak2, Atf4, Ddit3, Asns, Trib3, Nlrp1, Map1lc3b, Atg3, Atg5, Atg7, Atg10, Atg12, Atg16, Becn1, Gabarap, Gabarapl2, Sqstm1, Ddit4, Nupr1, Bcl2l11, Bbc3, Atf5, Pmaip1, Txnip* were tested using a corrected value of p≤0.05.

## Lentiviral packaging

Target plasmid was maintained in and purified from NEB 10-beta electrocompetent *E. coli* (New England Biolabs, C3020K). Lentivirus was packaged by plating $4×10^6$ HEK293T cells on 10 cm$^2$ and incubating cells overnight at 37°C. Cells were transfected with a plasmid mix consisting of 3.5 µg of the target plasmid, 6.0 µg psPAX2 (Addgene, 12260), and 3.0 µg pMD2.G (Addgene, 12259) using Lipofectamine 2000 (ThermoFisher Scientific, 11668019) in accordance with the manufacturer's instructions. Transfected HEK293T cells were incubated for 48 hr, before medium was collected, and centrifuged at 200×g for 5 mins. The resulting supernatant was filtered using a 0.45 µm filter before infection. CHO cells were plated at 20–30% confluency in and incubated overnight prior to infection. The packaged virus was applied to cells for 24 hr before the medium was exchanged for fresh medium. After another 24 hr of incubation, the medium was exchanged for fresh medium containing 10 µg/ml puromycin for 8 days.

## qPCR

gDNA and RNA were extracted from cells using the QIAamp DNA Kit (Qiagen, 51304) the Qiagen RNeasy Kit (Qiagen, 74104), respectively, in accordance with the manufacturer's protocol. For RNA extraction, QIAshredder homogenizer columns (Qiagen, 79654) were used to disrupt the cell lysates. cDNA was generated from RNA using Invitrogen SuperScript IV Reverse Transcriptase (Invitrogen, 18090200) and oligo(dT) primers. Each qPCR reaction was performed using SYBR Green Master I (Roche, 04707516001) on a Light Cycler 480 (Roche, 05015243001) using the recommended cycling conditions. Primers were designed to amplify amplicons 150–200 bp in size.

## Acknowledgements

We would like to thank the members of the Boeke and Wang labs for comments and discussion on the work and manuscript. RMM additionally thanks personal support from Xiaoyu Weng. Defense Advanced Research Projects Agency HR0011-17-2-0041 (HHW, JDB). National Institutes of Health / National Human Genome Research Institute RM1 HG009491 (JDB). National Institutes of Health / National Institute of Allergy and Infectious Diseases R01AI132403 (HHW). National Science Foundation MCB-2032259 (HHW). Burroughs Wellcome Fund PATH1016691 (HHW). Irma T Hirschl Trust (HHW). Dean's Fellowship from the Graduate School of Arts and Sciences of Columbia University (RMM).

## Additional information

### Competing interests

Jef D Boeke: is a Founder and Director of CDI Labs, Inc, a Founder of Neochromosome, Inc, a Founder and SAB member of ReOpen Diagnostics, and serves or served on the Scientific Advisory Board of the following: Sangamo, Inc, Modern Meadow, Inc, Rome Therapeutics, Sample6, Inc, Tessera Therapeutics, Inc and the Wyss Institute. Harris H Wang: H.H.W. is a scientific advisor to SNIPR Biome, Arranta Bio, Genus plc, Kingdom Supercultures, Fitibomics, and VecX Biomedicine and a scientific cofounder of Aclid. The other authors declare that no competing interests exist.

## Funding

| Funder | Grant reference number | Author |
| --- | --- | --- |
| Defense Advanced Research Projects Agency | HR011-17-2-0041 | Harris H Wang |
| National Human Genome Research Institute | RM1 HG009491 | Jef D Boeke |
| National Science Foundation | MCB-2032259 | Harris H Wang |
| Burroughs Wellcome Fund | PATH1016691 | Harris H Wang |
| Irma T. Hirschl Trust | | Harris H Wang |
| Dean's Fellowship from the Graduate School of Arts and Sciences of Columbia University | | Ross M McBee |
| National Institute of Allergy and Infectious Diseases | R01AI132403 | Ross M McBee |

The funders had no role in study design, data collection and interpretation, or the decision to submit the work for publication.

## Author contributions

Julie Trolle, Ross M McBee, Conceptualization, Formal analysis, Funding acquisition, Investigation, Visualization, Methodology, Writing – original draft, Writing – review and editing; Andrew Kaufman, Formal analysis, Investigation, Methodology, Writing – review and editing; Sudarshan Pinglay, Henri Berger, Sergei German, Liyuan Liu, Michael J Shen, Xinyi Guo, J Andrew Martin, Investigation, Methodology; Michael E Pacold, Drew R Jones, Supervision, Investigation, Methodology; Jef D Boeke, Harris H Wang, Conceptualization, Supervision, Funding acquisition, Visualization, Methodology, Writing – original draft, Project administration, Writing – review and editing

## Author ORCIDs

Julie Trolle (iD) http://orcid.org/0000-0002-2497-3531
Ross M McBee (iD) http://orcid.org/0000-0003-1926-2863
Michael E Pacold (iD) http://orcid.org/0000-0003-3688-2378
Jef D Boeke (iD) http://orcid.org/0000-0001-5322-4946
Harris H Wang (iD) http://orcid.org/0000-0003-2164-4318

## Decision letter and Author response

Decision letter https://doi.org/10.7554/eLife.72847.sa1
Author response https://doi.org/10.7554/eLife.72847.sa2

---

# Additional files

## Supplementary files

• Supplementary file 1. Table of minimal prototrophic pathways set for AA prototrophy in mammalian cells.

• Supplementary file 2. Protein sequences for each gene listed in *Supplementary file 1*.

• Supplementary file 3. Codon-optimized nucleotide sequence for each gene listed in *Supplementary file 1*.

• Supplementary file 4. Complete amino acid prototrophy pathway EC numbers.

• Transparent reporting form

## Data availability

Sequencing data generated for this study is deposited in the NCBI SRA at accession number PRJNA742028. Source data files have been provided for Figure 1 - figure supplement 1, Figure 1 - figure supplement 2D, Figure 2, Figure 2 - figure supplement 3, Figure 2 - figure supplement 4B, Figure 2 - figure supplement 5, Figure 2 - figure supplement 6, Figure 3, and Figure 3 - figure

supplement 1, Figure 4, Figure 4 - figure supplement 1, Figure 5, and Figure 5 - figure supplement 1.

The following dataset was generated:

| Author(s) | Year | Dataset title | Dataset URL | Database and Identifier |
|-----------|------|---------------|-------------|-------------------------|
| Trolle J | 2021 | Data from: Resurrecting essential amino acid biosynthesis in a mammalian cell | https://www.ncbi.nlm.nih.gov/bioproject/PRJNA742028 | NCBI BioProject, PRJNA742028 |

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
