## [Editor Report]

In this report, the authors devised synthetic genomic strategies to introduce essential amino-acid biosynthetic pathways into mammalian cells. While the functionalization of methionine, threonine, and isoleucine synthesis was unsuccessful, restoration of valine synthesis rendered mammalian cells partially independent of exogenous valine. Moreover, transcriptomes of the valine-prototrophic cell mirrored transcriptomes captured during recovery from valine deprivation in parental, valine-auxotrophic counterparts. Altogether, this work was found to be of substantial interest as it provides pioneering evidence that mammalian systems may be permissive to the restoration of essential amino acid biosynthetic pathways and is thus anticipated to have a broad impact in the fields of synthetic biology, biotechnology and beyond.

---

## [Decision Letter]

**Decision letter after peer review:**

Thank you for submitting your article "Resurrecting essential amino acid biosynthesis in a mammalian cell" for consideration by *eLife*. Your article has been reviewed by 3 peer reviewers, including Ivan Topisirovic as Reviewing Editor and Reviewer #1, and the evaluation has been overseen by Philip Cole as the Senior Editor. The following individual involved in review of your submission has agreed to reveal their identity: Ran Kafri (Reviewer #3).

Essential revisions:

1) CHO cells engineered to biosynthesize valine exhibited gradual increase in doubling time and reduction in valine production. Based on this, it was thought that more evidence is required to demonstrate that the introduction of valine biosynthetic pathway into CHO cells results in sustained proliferation and survival in the absence of valine supplementation. Accordingly, it was deemed that the authors should monitor long-term ability of engineered CHO cells to sustain valine production and proliferate in valine-free media.

2) Characterization of valine synthesizing CHO cells was found to be incomplete. To this end, monitoring flux via valine biosynthetic and degradation pathways, transcriptome and mTOR signaling at early and late time points was thought to be warranted.

3) Some technical issues were also observed. These include lack of clarity pertinent to the rationale behind using "conditioned-medium" in the experiments. Moreover, potential utilization of other sources of valine (e.g., proteins from the serum, autophagy) should be excluded.

4) More deliberation on the failure of the attempts to functionalize other essential amino acid biosynthesis in CHO cells should be considered.

5) The manuscript requires careful editing.

Reviewer #1 (Recommendations for the authors):

Specific comments:

– The extent of rescue of proliferation of CHO cells prototrophic for valine appears to be modest. It was appreciated that the latter cells survive in valine-free media, but it seems that their proliferation is significantly lower than in valine containing media. Moreover, it seems that after 6 passages only a fraction of the detected valine is synthesized de novo. Would this fraction further decrease in subsequent passages? Related to this, it is not clear what is the efficiency of valine biosynthesis in CHO cells vs. a prototrophic organism. Perhaps comparing the rates of valine synthesis in cell free extracts of CHO cells vs. those derived from a prototrophic organism may be helpful to address this.

– Another concern is that serum proteins, autophagy of proteasome may act as sources of valine, which should be excluded.

– Notwithstanding that monitoring steady-state mRNA levels was appreciated, further characterization of valine-prototrophic CHO cells appears to be warranted. This in particular relates to amino-acid sensing pathways (e.g., mTOR) that signal amino-acid availability to the proliferation/growth machinery.

– The authors should also elaborate on the failure to functionalize other tested pathways. Was this due to the altered stoichiometry and/or inadequate expression of the enzymes belonging to these pathways (which may occur post-transcriptionally)? Were the enzymes mislocalized? Are there other regulatory factors involved? Moreover, considering that the overarching tenet is that metazoans lost the ability to produce essential amino acids due to energetic restraints, it may be worthwhile noting that culturing conditions and potential differences in energy resources may impact on functionalization of essential amino acid biosynthetic pathways.

– Notwithstanding that this may be seen as too elaborate, including additional cell lines may be beneficial to exclude potential cell-line related biases.

*Reviewer #2 (Recommendations for the authors):*

The results put forth in this manuscript suggest the authors were marginally successful in introducing a valine biosynthetic pathway into CHO cells, but fall short of demonstrating a robust, self-sustaining engineered cell line under reasonable culture conditions. This milestone should be met prior to final acceptance at *eLife*. Additionally, the following revisions should be carried out prior to acceptance.

1) To substantiate the authors' major claims, follow up experiments demonstrating long-term homeostasis in the engineered cell line over multiple passages should be performed.

– It is possible residual valine from complete medium may help pCTRL and pMTIV cells survive at early timepoints. The authors should identify the timepoint at which pCTRL cells are no longer viable in dropout medium. The authors should then compare transcriptional profiles of the pMTIV cells at that timepoint to the that of pMTIV cells harvested at 4hr and 48hr.

– Similar to the point above, authors should include data examining pCTRL viability/ proliferation when fed with the conditioned medium regimen.

2) Metabolic flux should be more rigorously quantified for the pMTIV strain by performing the 13C6 tracer experiments and measuring 2-acetolactate, 2,3-dihydroxy-isovalerate, and 2-oxoisovalerate levels over time. Doing so may help identify key bottlenecks in the pathway. If a bottleneck can be identified, authors should attempt to make the pathway more efficient, either by modifying expression strategy of that enzyme or testing homologs from other hosts. The pathway should be optimized until the major revision 1 above is achieved.

3) Unsuccessful efforts to produce methionine, threonine, and isoleucine are distracting to the rest of the manuscript. For clarity, these sections should be de-emphasized in writing and figures for clarity.

4) Introducing heterologous pathways into new hosts can cause toxicity, especially when the donor and recipient are phylogenetically divergent. Authors should either experimentally test or comment on whether pathway intermediates/byproducts could have off-target effects.

*Reviewer #3 (Recommendations for the authors):*

The task that Wang et al., has undertaken is adventurous and the successful accomplishment can prove important in generating many unexpected insights once the CHO cell lines are shared with the wider community.

The authors have a few grammatical and spelling errors that should be corrected, including in the title "in a mammalian cell"  "in mammalian cells".

---

## [Author Response]

Essential revisions:1) CHO cells engineered to biosynthesize valine exhibited gradual increase in doubling time and reduction in valine production. Based on this, it was thought that more evidence is required to demonstrate that the introduction of valine biosynthetic pathway into CHO cells results in sustained proliferation and survival in the absence of valine supplementation. Accordingly, it was deemed that the authors should monitor long-term ability of engineered CHO cells to sustain valine production and proliferate in valine-free media.

We thank the reviewers for this feedback. We understand the core issue to be the reduction in doubling time shown for later time points in Figure 2F and the suggestion that this represents a time-dependent lag in growth rate due to cumulative insufficient valine production.

In response to this feedback, we set out to attain a consistent doubling time in the valine-free condition. We did so by increasing *ilvD* copy number in prototrophic pMTIV cells using lentiviral transduction thus generating a novel pMTIV-ilvD+ cell line that overexpresses *ilvD*-encoded dihydroxy-acid dehydratase (Figure 4A), a decision based on new information suggesting that bottlenecking was occurring at this step (see main point #2 below). Importantly, this dihydroxy-acid dehydratase overexpressing cell line was passaged 10 times in the absence of valine with a consistent average doubling time of 3.2 days. Doubling time remained consistent across the 39 days of culture and no medium conditioning was required (Figure 5).

Nonetheless, to alleviate concerns that the original prototrophic pMTIV cells were not able to sustain proliferation long-term in the absence of valine, we have also added additional evidence indicating that these cells retained valine prototrophy long-term:

1. A panel demonstrating the performance of an auxotrophic pCtrl cell line in the 50% conditioned F-12K medium originally used to culture pMTIV cells long-term (Figure 2 —figure supplement 5).

– pCtrl cells cultured in these conditions die off completely by day 8 having undergone zero passaging events

– Although pMTIV proliferation is slow at later time points in Figure 2F, we disagree with the suggestion that this represents a lack of prototrophy. Given the rapid death phenotype experienced by pCtrl cells, continued survival of pMTIV cells at late passages should instead be considered an indicator of sustained prototrophy.

2. Late time point transcriptomic data (Figure 3 —figure supplement 5) for pMTIV cells demonstrating partial rescue of nutritional starvation at day 29 in conditioned valine-free F-12K medium.

– Importantly, these data demonstrate that pMTIV cells at day 29 in the absence of valine are more similar to pCtrl cells cultured in complete medium than pCtrl cells cultured in the absence of valine for just 48 h.

3. Both early and late time point ^13^C-tracing data (Figure 4E, Figure 4 —figure supplement 1D) for pMTIV and pMTIV-ilvD+ cells demonstrated ongoing valine biosynthesis at late time points.

– If cells were increasingly relying on supplementation of valine from an exogenous source at later time points, we would expect to see a reduction in ^13^C-labeling of valine towards later time points. However, no such trend was evident in either pMTIV or pMTIV-ilvD+ cells.

2) Characterization of valine synthesizing CHO cells was found to be incomplete. To this end, monitoring flux via valine biosynthetic and degradation pathways, transcriptome and mTOR signaling at early and late time points was thought to be warranted.

We thank the reviewers for these comments. We have added a figure highlighting mTOR signaling differences in pMTIV and pCtrl cells at 48 h valine starvation, even though no clear signatures of mTOR activation could be detected (Figure 3 —figure supplement 4). We have also added a new supplemental figure showing transcriptomic analysis of cells grown long-term (5 passages, 29 days) in conditioned valine-free F-12K medium (Figure 3 —figure supplement 5).

Additionally, we were able to gain insight into flux through the pathway with ^13^C-tracing. No signal could be detected for pyruvate, 2-acetolactate or 2-oxoisovalerate; however we were able to specifically detect pathway intermediate 2,3-dihydroxy-isoverate and have added a panel to reflect this (Figure 3 —figure supplement 1F).

It was unclear whether the detected 2,3-dihydroxy-isoverate represented a true pathway bottleneck. In order to test whether this was the case, we introduced extra copies of the downstream *ilvD* gene encoding the dihydroxy-acid dehydratase enzyme, by lentiviral transduction. In doing so, we generated the pMTIV-ilvD+ cell line, the growth of which outperformed the original pMTIV cell line significantly in valine-free RPMI medium, suggesting that *ilvD* expression levels led to inefficient flux through the pathway in valine starvation conditions. The newly generated pMTIV-ilvD+ cell line grew at an average doubling time of 3.2 days over 39 days in unconditioned valine-free RPMI medium compared to 1.0 days in the complete RPMI medium condition (Figure 5).

3) Some technical issues were also observed. These include lack of clarity pertinent to the rationale behind using "conditioned-medium" in the experiments. Moreover, potential utilization of other sources of valine (e.g., proteins from the serum, autophagy) should be excluded.

We apologize for the lack of clarity surrounding the use of conditioned medium and thank the reviewers for bringing this to our attention. We have added a panel demonstrating the utility of using conditioned medium in culturing pMTIV cells in the absence of valine (Figure 2 —figure supplement 5B). Furthermore, since we are now able to culture the new pMTIV-ilvD+ cells without conditioning, this critique may be considered moot.

The rationale behind the conditioned medium approach assumes amino acid transport in and out of cells to be adapted for a specific equilibrium of intracellular/extracellular valine concentration. When culturing prototrophic cells in valine-free medium conditions, extracellular valine concentrations will be minimal, forcing cells to secrete valine until the appropriate equilibrium has been met. By ‘conditioning’ the medium with prototrophic pMTIV cells ahead of time, we avoid disadvantaging the cells by requiring them to supply yet another valine-depleted environment with valine before retaining valine for anabolic purposes.

Examples supporting this rationale can be found in the literature. For example, in a 1962 publication by Eagle and Piez^1^ it was demonstrated that there is a population-dependent requirement of cultured cells for metabolites that are otherwise considered non-essential. For instance, serine was required for growth when cells were cultured at low cell densities. In the words of the authors “the critical population density was that which was able to ‘condition’ the medium i.e. to build up a concentration in equilibrium with the minimum effective intracellular level, before the cells died of the specific deficiency”.

Figure 2 —figure supplement 5B further supports this explanation by illustrating that the positive effect from medium conditioning cannot be recapitulated if the medium is conditioned with pCtrl cells, which excludes the possibility of cell debris or other effects from medium conditioning conferring the positive benefit. It would therefore indicate that the benefit to cells that is derived from pMTIV medium conditioning is likely specifically caused by the valine synthesized and secreted by these cells.

With regard to potential utilization of other sources of valine, we agree with the reviewers’ concern and we have added a supplementary figure illustrating the amino acid content in the dialyzed fetal bovine serum used for dropout media in this study (Figure 2 —figure supplement 3). The serum was analyzed for the presence of 15 amino acids including valine, which was found to be present at 9.25 μM in the dialyzed fetal bovine serum used to supplement the dropout media. For context, when used at a 10% concentration, this constitutes a mere 0.47% contribution to valine content for a standard complete F-12K medium formulation (not supplemented with FBS) or 0.54% for RPMI 1640 (not supplemented with FBS), neither of which is sufficient to rescue starvation effects as evidenced by the rapid death seen for pCtrl cells when cultured in valine-free medium (Figure 2C, Figure 2E).

Regarding autophagy, if such an effect would significantly alter the outcome of cells, this would not be specific to our engineered cells and any rescue effects thereof should be apparent for pCtrl cells as well, which was shown not to be the case (Figure 2C, Figure 2E, Figure 2 —figure supplement 5).

4) More deliberation on the failure of the attempts to functionalize other essential amino acid biosynthesis in CHO cells should be considered.

Given the success with valine, we feel it appropriate to outline these results on their own terms. However, we agree that it would be beneficial to additionally discuss other efforts.

We initially began our experimentation by designing an all-in-one construct that would introduce (a) isoleucine and valine biosynthesis using a shared 4-gene pathway (b) threonine biosynthesis by driving a typically degradative enzyme in reverse, and (c) rescue of methionine auxotrophy by bridging a gap in the sulfur shuttle. The all-in-one format using 2A ribosome-skipping peptide sequences served to free up the limited number of available mammalian regulatory elements for potential addition of other pathway functionalities as well as to minimize the number of genes introduced and by extension the cost of DNA synthesis. In particular, the gene choices made in the attempts to achieve (b) and (c) were optimistic and made in the interest of optimizing pathway number per DNA length.

While the valine pathway in theory is able to conduct isoleucine biosynthesis activity as well, the choice of an *E. coli*-derived *ilvN/B* acetolactate synthase to catalyze the initial pathway step may make it a valine biosynthetic pathway in practice as this enzymatic complex does not specifically favor isoleucine biosynthesis over valine biosynthesis given similar substrate availabilities like other *E. coli* isozymes (*ilvG/M, ilvI/H*) might^2^. This may be necessary for meaningful isoleucine biosynthetic functionality but in addition, isoleucine biosynthesis additionally requires the presence of 2-oxobutanoate, which is not as involved in core metabolism as pyruvate and therefore is presumably found at much lower concentrations in cells. Furthermore, the *ilvN/B*-encoded acetolactate synthase used here is known to be negatively feedback inhibited by all three branched-chain amino acids but most strongly by isoleucine. We have added a panel (Figure 4 —figure supplement 1B) demonstrating increased proliferative ability of pMTIV cells in valine-free RPMI medium at a reduced (0.06 mM instead of the standard 0.38 mM) isoleucine concentration. Collectively, these two biochemical constraints may limit the cells’ ability to produce self-sufficient quantities of isoleucine.

In the case of threonine, we attempted to opportunistically take advantage of the bidirectionality of a typically degradative enzyme, *ltaE*. However, this failed to rescue threonine auxotrophy, presumably because the mammalian metabolic equilibrium did not favor the reverse enzymatic reaction as intended.

In the case of methionine, rescue of biosynthesis was attempted by allowing for interconversion of cystathionine and homocysteine. Methionine is synthesized in mammalian cells from homocysteine, and we reasoned that increasing levels of cystathionine by introduction of *E. coli*-derived *metC* would increase levels of homocysteine, which might increase cell viability in methionine-free conditions. However, cystathionine biosynthesis in *E. coli* and mammalian cells are divergent processes requiring different starting substrates. Whereas *E. coli* synthesizes cystathionine from cysteine and succinyl-homoserine, mammalian cells synthesize cystathionine from serine and homocysteine. Introducing *metC* into a mammalian metabolic context therefore bridges a gap that is incompatible with the evolutionary developments of the past hundreds of millions of years, resulting in a circular pathway unlikely to produce significant quantities of methionine, which was confirmed empirically in our functional assay.

We would like to highlight to the reviewers that additional work is ongoing to rescue yet other essential amino acids, as well as our call for a wider community focus on such projects.

5) The manuscript requires careful editing.

We have performed additional editing of the text to remove unfortunate errors.

*Reviewer #1 (Recommendations for the authors):*

Specific comments:– The extent of rescue of proliferation of CHO cells prototrophic for valine appears to be modest. It was appreciated that the latter cells survive in valine-free media, but it seems that their proliferation is significantly lower than in valine containing media. Moreover, it seems that after 6 passages only a fraction of the detected valine is synthesized de novo. Would this fraction further decrease in subsequent passages? Related to this, it is not clear what is the efficiency of valine biosynthesis in CHO cells vs. a prototrophic organism. Perhaps comparing the rates of valine synthesis in cell free extracts of CHO cells vs. those derived from a prototrophic organism may be helpful to address this.

We would like to clarify that the metabolomics data presented in the manuscript describes a separate experiment from the long-term culture experiments, and were collected after 3 passages or 12 days in unconditioned valine-free RPMI medium containing ^13^C-glucose and ^13^C-sodium pyruvate (Figure 3 —figure supplement 1A).

To measure valine biosynthesis past the 3^rd^ passage as suggested, we set out to perform an additional metabolomics analysis looking at ^13^C-valine levels – this time over a longer time period. In this time course, ^13^C-glucose replaced its ^12^C counterpart in the valine-free RPMI medium formulation as before; however the spiked in sodium pyruvate was not ^13^C-labeled in this follow-up experiment due to limited reagent availability during the COVID-19 pandemic. This is important to note as it follows that the expected ^13^C-labeling outcome is different. When spiking in ^12^C-sodium pyruvate, a subset of the M+0 valine detected in cells is likely to have been synthesized endogenously (from ^12^C pyruvate) as is all of the detected M+2, M+3, and M+5 valine (Figure 4 —figure supplement 1A). This is in contrast to the original experiment in which only ^13^C sources of glucose and pyruvate were spiked in. In that case, nearly all valine was expected to be M+5 if synthesized endogenously (Figure 3 —figure supplement 1A).

However it was unclear whether we would be able to culture cells long-term in unconditioned RPMI rather than in 50% conditioned F-12K medium which was originally used for long-term culture in valine-free conditions. We were unable to continue using the 50% conditioned F-12 as (a) for the purposes of ^13^C-tracing, a conditioned medium regimen would complicate downstream analysis, and (b) we did not have access to a glucose-depleted F-12K powder base that would allow for full replacement of ^12^C glucose/pyruvate with its ^13^C equivalent.

In anticipation that cells might not perform well in unconditioned medium and in the new RPMI context, we therefore attempted to take measures to lose fewer cells to the harsh effects of passaging by culturing cells on plates coated with 0.1% gelatin thereby enabling metabolomic analysis of cultured in near-minimal valine conditions. While it in theory was possible that cells were consuming valine derived from the 0.1% gelatin coated on plates, any potential gelatin-derived valine would not be ^13^C-labeled and should not influence information gained regarding ^13^C-labeling levels. Furthermore, we later cultured cells long-term in unconditioned valine-free RPMI on plates not coated with 0.1% gelatin (Figure 5) and found growth in this condition to be no slower than growth on plates coated with 0.1% gelatin (Figure 4 —figure supplement 1C) suggesting that it is unlikely that cells are using valine derived from gelatin for anabolic purposes.

Over the time course, we saw no clear downward trend in the proportion or overall content of ^13^C-labeled valine over the time course for pMTIV or pMTIV-ilvD+ cells, demonstrating that the cells were not reducing de novo synthesis of valine over time (Figure 4 —figure supplement 1D, Figure 4E). In fact, for pMTIV-ilvD+ cells, ^13^C-valine content was higher than ^12^C-valine content at every time point past 4 days of culture in the valine-free ^13^C-labeled RPMI medium demonstrating that cells primarily rely on de novo synthesis of valine.

In pMTIV cells, ^13^C-valine content was lower than ^12^C-valine content on days 14 and 24 while the opposite was true on days 2, 4, 12, and 18, demonstrating that the ^13^C content of the cells was not on a downward trend but rather fluctuated up and down. This may reflect an inability to adequately respond to valine demands given inefficient flux through the pathway.

– Another concern is that serum proteins, autophagy of proteasome may act as sources of valine, which should be excluded.

We have addressed this point above in our response to Essential Revision 3.

– Notwithstanding that monitoring steady-state mRNA levels was appreciated, further characterization of valine-prototrophic CHO cells appears to be warranted. This in particular relates to amino-acid sensing pathways (e.g., mTOR) that signal amino-acid availability to the proliferation/growth machinery.

We have addressed this point above in our response to Essential Revision 2.

– The authors should also elaborate on the failure to functionalize other tested pathways. Was this due to the altered stoichiometry and/or inadequate expression of the enzymes belonging to these pathways (which may occur post-transcriptionally)? Were the enzymes mislocalized? Are there other regulatory factors involved? Moreover, considering that the overarching tenet is that metazoans lost the ability to produce essential amino acids due to energetic restraints, it may be worthwhile noting that culturing conditions and potential differences in energy resources may impact on functionalization of essential amino acid biosynthetic pathways.

We thank the reviewer for these insights. We address this point above in our response to Essential Revision 4.

– Notwithstanding that this may be seen as too elaborate, including additional cell lines may be beneficial to exclude potential cell-line related biases.

We agree with the reviewer on this point and have already begun efforts to test our pathways in other cell lines, such as HEK cells, but we believe these data are too preliminary to be included at this time, and are beyond the scope of this contribution.

Reviewer #2 (Recommendations for the authors):The results put forth in this manuscript suggest the authors were marginally successful in introducing a valine biosynthetic pathway into CHO cells, but fall short of demonstrating a robust, self-sustaining engineered cell line under reasonable culture conditions. This milestone should be met prior to final acceptance at eLife. Additionally, the following revisions should be carried out prior to acceptance.

We undertook a painstaking and extensive effort to demonstrate robust and self-sustaining valine-free growth over 39 days. This was achieved by increasing *ilvD* (encoding a dihydroxy-acid dehydratase enzyme) copy number in response to detecting a potential pathway bottleneck in 2,3-dihydroxy-isovalerate. By increasing the efficiency of the final step in the introduced biosynthetic pathway, doubling time was reduced to a relatively consistent 3.2 days per doubling across 39 days in unconditioned RPMI medium supplemented with dialyzed FBS (Figure 5).

Major revisions:1) To substantiate the authors' major claims, follow up experiments demonstrating long-term homeostasis in the engineered cell line over multiple passages should be performed.o It is possible residual valine from complete medium may help pCTRL and pMTIV cells survive at early timepoints. The authors should identify the timepoint at which pCTRL cells are no longer viable in dropout medium. The authors should then compare transcriptional profiles of the pMTIV cells at that timepoint to the that of pMTIV cells harvested at 4hr and 48hr.– Similar to the point above, authors should include data examining pCTRL viability/ proliferation when fed with the conditioned medium regimen.

We have included a figure describing pCtrl inviability using the 50% conditioned F-12K medium regimen (Figure 2 —figure supplement 5A).

With regard to pMTIV cells surviving on residual valine in later passages, we believe this is unlikely given that Figure 2E and Figure 2 —figure supplement 5A show that pCtrl cells typically die before the first passaging event in valine-free medium – after approximately 6 days in unconditioned valine-free F-12K medium or approximately 8 days in 50% conditioned valine-free F-12K medium.

While it was possible to include an 8 day timepoint for collecting transcriptomic pMTIV samples we originally did not do so as there is no suitable control for analyzing such samples. Nonetheless, we had previously collected RNA in duplicate samples from a late time point and as such have included transcriptomic data in a new figure for samples cultured in conditioned valine-free F-12K medium over 29 days or 5 passages i.e. well past the point of pCtrl inviability (Figure 3 —figure supplement 5)**.** Evidently, these cells more closely resemble healthy cells cultured in complete medium than do pCtrl cells cultured in valine-free F-12K for just 48 h.

2) Metabolic flux should be more rigorously quantified for the pMTIV strain by performing the 13C6 tracer experiments and measuring 2-acetolactate, 2,3-dihydroxy-isovalerate, and 2-oxoisovalerate levels over time. Doing so may help identify key bottlenecks in the pathway. If a bottleneck can be identified, authors should attempt to make the pathway more efficient, either by modifying expression strategy of that enzyme or testing homologs from other hosts. The pathway should be optimized until the major revision 1 above is achieved.

This excellent reviewer suggestion led to an important new finding – the specific accumulation of an intermediate suggesting a pathway bottleneck. We explored the presence of pathway intermediates in our original ^13^C-tracing experiment. The only pathway intermediate detected by metabolomics analysis was 2,3-dihydroxy-isovalerate, suggesting potential bottlenecking at this step. We thus optimized our pMTIV cell line by introducing extra copies of the downstream dihydroxy-acid dehydratase-encoding *ilvD* gene by lentiviral transduction thus generating a novel pMTIV-ilvD+ cell line (Figure 4A). This resulted in cells that double on average every 3.2 days across 39 days in unconditioned valine-free RPMI medium (Figure 5). Doubling time was relatively consistent across the entire time course and long-term ^13^C-tracing demonstrated that valine content as well as de novo valine biosynthesis in the novel pMTIV-ilvD+ cell line was stable across 36 days in unconditioned valine-free RPMI medium compared to pMTIV cells which exhibited fluctuating valine levels (Figure 4 —figure supplement 1D, Figure 4E). This data demonstrates long-term homeostasis and robust growth under reasonable culturing conditions.

3) Unsuccessful efforts to produce methionine, threonine, and isoleucine are distracting to the rest of the manuscript. For clarity, these sections should be de-emphasized in writing and figures for clarity.

We believed it would be misleading to describe our efforts to rescue valine biosynthesis alone. On the suggestion of Reviewer #1 above as well as the suggestion outlined in Essential Revision #4 sections have been adjusted but not removed.

4) Introducing heterologous pathways into new hosts can cause toxicity, especially when the donor and recipient are phylogenetically divergent. Authors should either experimentally test or comment on whether pathway intermediates/byproducts could have off-target effects.

The potential toxicity of valine pathway intermediates (or perhaps toxic products from non-specific enzymatic activity) is certainly an interesting question, given the introduction of a pathway sourced from a distantly removed species. However, we saw no signs of metabolic stress in cells harboring the pathway when grown on complete media, either by growth rate (Figure 2D) or in the transcriptomic data (Figure 3D, Figure 3 —figure supplement 2). The introduction of the pathway therefore does not appear to be a significant stressor to the cells. However, it remains to be seen if this will true for all essential amino acids, particularly as we look to introduce the more complex pathways.

Reviewer #3 (Recommendations for the authors):The task that Wang et al., has undertaken is adventurous and the successful accomplishment can prove important in generating many unexpected insights once the CHO cell lines are shared with the wider community.

We thank the reviewer for the vote of confidence, and share their excitement for the insights this work might enable.

The authors have a few grammatical and spelling errors that should be corrected, including in the title “in a mammalian cell” “ "in mammalian cells".

We apologize for these errors, and have performed additional proofing.

References

1. Eagle, H. and Piez, K. The population-dependent requirement by cultured mammalian cells for metabolites which they can synthesize. *Journal of Experimental Medicine* 116, 29–43 (1962).

2. Barak Z, Chipman D M, and Gollop N. Physiological implications of the specificity of acetohydroxy acid synthase isozymes of enteric bacteria. *Journal of Bacteriology* 169, 3750–3756 (1987).

3. Mitchell Leslie A. *et al.,* Synthesis, debugging, and effects of synthetic chromosome consolidation: synVI and beyond. *Science* 355, eaaf4831 (2017).

4. Richardson Sarah M. *et al.,* Design of a synthetic yeast genome. *Science* 355, 1040–1044 (2017).